# Genomic Determinants of Knee Joint Biomechanics: An Exploration into the Molecular Basis of Locomotor Function, a Narrative Review

Georgian-Longin Iacobescu [1,2], Loredana Iacobescu [1,2,*], Mihnea Ioan Gabriel Popa [1,2], Razvan-Adrian Covache-Busuioc [1], Antonio-Daniel Corlatescu [1] and Catalin Cirstoiu [1,2,*]

[1] Orthopaedics and Traumatology Department, "Carol Davila" University of Medicine and Pharmacy, 050474 Bucharest, Romania; georgianyak@yahoo.com (G.-L.I.); mihnea.popa@umfcd.ro (M.I.G.P.); razvan-adrian.covache-busuioc0720@stud.umfcd.ro (R.-A.C.-B.); antonio.corlatescu0920@stud.umfcd.ro (A.-D.C.)

[2] University Emergency Hospital, 050098 Bucharest, Romania

* Correspondence: loredana.gheorghiu@drd.umfcd.ro (L.I.); catalin.cirstoiu@umfcd.ro (C.C.)

**Abstract:** In recent years, the nexus between genetics and biomechanics has garnered significant attention, elucidating the role of genomic determinants in shaping the biomechanical attributes of human joints, specifically the knee. This review seeks to provide a comprehensive exploration of the molecular basis underlying knee joint locomotor function. Leveraging advancements in genomic sequencing, we identified specific genetic markers and polymorphisms tied to key biomechanical features of the knee, such as ligament elasticity, meniscal resilience, and cartilage health. Particular attention was devoted to collagen genes like COL1A1 and COL5A1 and their influence on ligamentous strength and injury susceptibility. We further investigated the genetic underpinnings of knee osteoarthritis onset and progression, as well as the potential for personalized rehabilitation strategies tailored to an individual's genetic profile. We reviewed the impact of genetic factors on knee biomechanics and highlighted the importance of personalized orthopedic interventions. The results hold significant implications for injury prevention, treatment optimization, and the future of regenerative medicine, targeting not only knee joint health but joint health in general.

**Keywords:** knee biomechanics; osteoarthritis; COL1A1; COL5A1; extracellular matrix; proteoglycans

## 1. Introduction

The escalating global prevalence of aging and obesity has intensified the focus on research pertaining to human motion dysfunction. Within this context, the knee joint, serving as the principal motor joint of the lower limb, emerges as the most susceptible to vulnerabilities [1]. Knee disorders, prevalent among these demographics, profoundly affect not only their functional mobility but also their mental well-being [2]. The knee's functional responsibilities encompass supporting body weight, facilitating lower limb swing, and absorbing impact shocks [3]. Movement biomechanics, a critical subdivision of biomechanics, delves into the synergistic functionality of bones, muscles, ligaments, and tendons during diverse human movements [4]. This intricate interplay enables the knee to endure significant forces during routine activities [1]. Consequently, there exists an imperative need for in-depth studies on the movement biomechanics of both healthy and impaired knee joints, aiming to enhance or rehabilitate human locomotive abilities.

The meniscus, a fibrocartilaginous structure, is pivotal in regulating the intricate biomechanics of the knee joint [5]. Its effective performance is contingent upon the specific composition and structural organization of its extracellular matrix (ECM) [6].

Articular cartilage, a critical component in the diarthrodial joints of skeletally mature individuals, is characterized by its avascular and aneural nature, serving essential roles

in stress distribution and facilitating low-friction joint movement [7,8]. Microscopically, this specialized tissue exhibits a highly organized structure stratified into four distinct zones: superficial, middle, deep, and calcified cartilage zones [9,10]. These zones display variation in their biochemical composition; notably, the superficial zone is distinguished by its elevated collagen concentration and cellularity, coupled with a reduced proteoglycan presence [11]. Contrastingly, proteoglycan content predominantly peaks in the middle zone.

Given its avascular nature, articular cartilage relies primarily on diffusion for the transport of nutrients, waste products, signaling molecules, and oxygen. This transport process is governed by two fundamental parameters: the diffusion coefficient and the partition coefficient. The diffusion coefficient, which quantifies the rate at which substances are transported via random molecular movement, is affected by the structural arrangement of collagen and proteoglycans within the cartilage matrix. Empirical studies on porcine cartilage have demonstrated that the diffusion coefficient for a 70 kDa dextran in the superficial zone is approximately 30% of that in the middle or deep zones [12]. Furthermore, the diffusivity of uncharged molecules, such as dextrans, is inversely related to the proteoglycan content and has been observed to diminish under conditions of static compression [13]. Meanwhile, the partition coefficient describes the relative concentration of a molecule within the cartilage's interstitial space compared to its concentration in a free solution. This parameter is integral to understanding the molecular behavior within the cartilaginous matrix and its implications for cartilage health and pathology.

The extracellular matrix (ECM) constitutes a dynamic, cell-secreted molecular network that not only provides structural integrity to tissues but also actively influences cellular behavior, including proliferation and differentiation. This intricate matrix serves as the architectural foundation within tissues, shaping the morphology of cells and maintaining the overall form of organs [14]. The diversity of the ECM is evident through the varying types of matrix molecules secreted by different cell types, with the composition and quantity of these molecules undergoing alterations across developmental stages. In the context of cartilage, the ECM is primarily composed of two fundamental components that define its mechanical and physical attributes: a collagenous network and proteoglycans, predominantly aggrecan. The collagen framework is pivotal in imparting tensile strength to the cartilage matrix, whereas proteoglycans are crucial for inducing osmotic swelling and bestowing elastic properties to the cartilage tissue [14].

The metamorphosis of cartilage into bone is a multifaceted process intricately involving various ECM components. This transformation is governed by a series of events that are essential for the structural and functional evolution of these tissues. Maintaining homeostasis in cartilage and bone involves complex mechanisms that regulate the turnover and remodeling of the ECM. This regulation is not confined to the mechanical aspects but extends to the production of local factors, inflammatory mediators, and matrix-degrading enzymes by the ECM-resident cells in both bone and cartilage. Furthermore, the turnover and degradation of both normal and pathological matrices are contingent upon the responses of local cells to autocrine and paracrine signals that drive anabolic and catabolic pathways. These cellular responses are critical in dictating the dynamic equilibrium of the ECM, influencing both its structural integrity and functional capacity in health and disease states [15,16].

## 2. Materials and Methods

### 2.1. Study Design and Methodology

We conducted a comprehensive search of the PubMed database for the most relevant articles regarding knee genomic determinants, biomechanics, and collagen genes and functions. For the search formula, we used the following terms: "knee biomechanics", "meniscus degeneration", "genetics of osteoarthritis", "collagen genes", "extracellular matrix", and "proteoglycans". Initially, the PubMed database showed 487 results. In this study, a thorough evaluation of article titles was conducted to ascertain the inclusion of

at least one relevant search term. Articles that did not satisfy the inclusion criteria or focused on subjects divergent from the genomic determinants and biomechanics of the knee joint were methodically excluded. Consequently, the final analysis incorporated 136 studies. The research was conducted from a holistic viewpoint, concentrating on the nuanced interplay between knee biomechanics and genomic factors. The approach was all-encompassing, endeavoring to comprehensively decipher the intricate relationship between genetic influences and the mechanical functionality of the knee joint.

### 2.2. Objectives

The main objective of this review is to thoroughly investigate and elucidate the molecular and genetic foundations that can influence the biomechanical characteristics of the knee joint, including the identification and analysis of specific genetic markers and polymorphisms that are directly linked to crucial biomechanical aspects of the knee, such as ligament elasticity, meniscal resilience, and cartilage integrity. Through this study, we seek to deepen the understanding of the interplay between genetics and knee biomechanics, thereby contributing to the advancement of injury prevention, more effective treatment approaches, and the innovation of regenerative medical practices that target not only the health of the knee joint but also the overall health of human joints. Another important objective is to provide an in-depth and meticulous review of the existing knowledge in this specific field, highlighting key research initiatives, methodologies, and findings. The categorization within this section is methodically organized based on thematic elements, distinct research areas, or particular aspects of the subject matter.

### 3. Collagen and the Knee's Structural Integrity

Collagens represent the predominant protein class in mammals, accounting for approximately 30% of the total protein mass and serving as a fundamental component of the extracellular matrix (ECM). This protein family encompasses 28 distinct subtypes, ranging from type I to type XXVIII, with type I collagen being the most prevalent, constituting about 90% of the body's collagen [17]. Each collagen subtype is capable of forming either a homotrimer or a heterotrimer, which are composed of three alpha chains.

The biosynthesis of these alpha chains initiates with the formation of procollagen, characterized by the presence of N-terminal and C-terminal propeptides. This procollagen undergoes a conformational change to form a triple helix structure within the cytoplasm. Following secretion from the cell, a critical post-translational modification occurs wherein both the N- and C-terminal propeptides are excised by specific proteinases. This process facilitates the subsequent crosslinking and assembly of the collagen molecules into fibrils, which is pivotal for their functional integration into the ECM [18,19]. This intricate process of collagen synthesis and assembly is crucial for maintaining the structural integrity and functional properties of the ECM in various tissues.

### 3.1. COL1A1

Collagen, a principal constituent of the extracellular matrix (ECM), is indispensable for the normal functioning of tissues, playing a pivotal role in sustaining the stability and structural integrity of both tissues and organs. Among its various forms, collagen type I alpha 1 chain (COL1A1), a key element of type I collagen, is extensively distributed across the body, particularly within the interstitial spaces of parenchymal organs and connective tissues. This wide distribution underscores its significance in facilitating tissue development and maintaining homeostasis [20].

Ligaments and tendons, key structural components in the human body, are composed of collagenous bands of fibrils, which include a variety of collagen types, proteoglycans, and glycoproteins [21]. Among these, Type I collagen is the predominant protein, accounting for 70–80% of the dry weight of ligaments [22]. This specific collagen molecule is a heterotrimer, composed of two alpha-1 (I) chains and one alpha-2 (I) chain, with the genes COL1A1 and COL1A2 responsible for encoding these chains, respectively [23].

The COL1A1 gene, in particular, has garnered significant attention due to its association with various medical conditions. Notably, the Sp1 binding site polymorphism within COL1A1 has been linked to an increased risk of cruciate ligament ruptures [24]. Additionally, mutations in the COL1A1 gene have been identified as causal factors in monogenic connective tissue disorders such as osteogenesis imperfecta and Ehlers–Danlos syndrome [23]. This functional Sp1 binding site polymorphism has also been associated with multifactorial disorders, including osteoporotic fractures, variations in bone mineral density, osteoarthritis, myocardial infarction, lumbar disc disease, and stress urinary incontinence [25–28].

It has been proposed that a specific substitution within the intronic Sp1 binding site (referred to as a GRT substitution) enhances the binding affinity for the transcription factor Sp1, leading to an upregulation in COL1A1 gene expression [26]. Given the critical role of Type I collagen in the structural integrity of ligaments, the connection between the COL1A1 gene and anterior cruciate ligament (ACL) ruptures, a common and severe injury among athletes, is particularly noteworthy and warrants further investigation. Another common athlete injury that should also be taken into consideration for further studies is the rotator cuff tear, which is considered to be the main shoulder injury [29].

In their research, Khoschnau et al. discovered an association between the risk of cruciate ligament ruptures and shoulder dislocations and a specific polymorphism in the COL1A1 gene. Notably, individuals possessing the rare SS genotype, which had a 4% prevalence in their study, were less frequently observed in the injured group, suggesting a significantly lower risk of these soft tissue injuries. The study highlights the complex role of COL1A1, a gene encoding for a common collagen type found in bones, ligaments, and joint cartilage. Regulation of COL1A1 transcription varies, with one key regulatory site being the Sp1 transcription factor-binding site. Functional analysis revealed that the s allele of the Sp1 polymorphism, linked to osteoarthritis and osteoporosis, is associated with enhanced DNA-protein binding, increased transcription, and higher production of collagen type I$\alpha$1 mRNA and protein. The Sp1 polymorphism in the COLIA1 gene, found on chromosome 17, is autosomal and exhibits a recessive pattern of inheritance. It is distributed equally among women and men [24,30].

Type I collagen, composed of two $\alpha$1 chains and one $\alpha$2 chain encoded by genes on chromosomes 17 and 7, is known for its high tensile strength and resistance to most proteases [31]. While essential for providing mechanical strength to tissues—as seen in osteogenesis imperfecta [32]—its abnormal accumulation can contribute to fibrotic diseases, underscoring the delicate balance of collagen in tissue health and disease.

In the study led by Posthumus et al., three key findings emerged regarding ACL (anterior cruciate ligament) ruptures and genetic factors. First, the rare TT genotype of the COL1A1 Sp1 binding site polymorphism was significantly less common among individuals who experienced ACL ruptures, suggesting that this genotype might offer some protection against such injuries. Second, the study found that individuals with an ACL rupture were over four times more likely to have a family history of ligament injuries compared to control participants. Third, it was observed that the majority of ACL ruptures in their study group occurred due to non-contact events, and the distribution of genotypes in this subgroup was consistent with those in subgroups sustaining indirect and direct injuries [33].

The synthesis of data from independent studies made in South Africa [33,34] and Sweden [24] strengthens the idea of an association between a particular genotype and the risk of acute soft tissue ruptures. Confirmation of these findings could significantly impact clinical practice, as it suggests COL1A1 has a major role in maintaining the structural integrity of soft tissue.

COL1A1/2 osteogenesis imperfecta (COL1A1/2-OI), a genetic disorder primarily characterized by fractures resulting from minimal or no trauma, presents with a spectrum of clinical manifestations. These manifestations range from perinatal lethality to severe skeletal deformities, mobility impairments, and notably shortened stature, extending to individuals who are nearly asymptomatic with only a mild predisposition to fractures. This

spectrum also includes normal dentition, stature, and life expectancy. The disorder can also feature variable dentinogenesis imperfecta (DI) and, in adults, hearing loss. Fractures in COL1A1/2-OI, more commonly occurring in the extremities, are a hallmark of the condition. DI, when present, is typified by teeth that are gray or brown, possibly translucent, and prone to rapid wear and breakage. The classification of COL1A1/2-OI has been refined into four primary types, differentiated by their clinical and radiographic presentations. This categorization aids in the prognostication and management of affected individuals. These types are:

1. Classic non-deforming OI with blue sclerae (formerly known as OI type I);
2. Perinatally lethal OI (previously OI type II);
3. Progressively deforming OI (formerly OI type III);
4. Common variable OI with normal sclerae (formerly OI type IV).

The diagnosis of COL1A1/2-OI is confirmed in a proband through the identification of a heterozygous pathogenic or likely pathogenic variant in either the COL1A1 or COL1A2 gene, established via molecular genetic testing. This diagnostic approach underscores the genetic basis of the disorder and facilitates precise medical intervention, genetic counseling, and management strategies tailored to the specific type of OI diagnosed [35]. COL1A1 is a gene, usually with heightened expression in multiple cancer types, that can potentially influence key cellular processes like cell proliferation, metastasis, apoptosis, and cisplatin resistance. Its elevated expression correlates with a poorer prognosis in cancer patients, indicating its role in cancer progression. However, the specific function of COL1A1 as a cancer-promoting factor in particular tumors remains unclear. Furthermore, the expression levels and mechanisms of action of the COL1A1 protein vary across different tumor types [36,37].

COL1A1/2-Osteogenesis Imperfecta (COL1A1/2-OI) follows an autosomal dominant inheritance pattern. The likelihood of this condition being a simplex case (a single occurrence in a family) varies with the disease's severity. About 60% of mild OI cases are simplex, whereas almost all cases of the more severe, progressively deforming, or perinatally lethal OI are simplex, often due to a de novo pathogenic variant or a variant inherited from a parent with somatic and/or germline mosaicism. Up to 16% of families may have parental somatic and/or germline mosaicism. Children of an individual with dominantly inherited COL1A1/2-OI have a 50% chance of inheriting the causative variant and potentially developing OI symptoms. Prenatal testing for at-risk pregnancies is possible through molecular genetic testing if the causative COL1A1 or COL1A2 variant is identified in a family member. Additionally, prenatal ultrasound examinations in experienced centers can help diagnose lethal and severe forms of OI before 20 weeks of gestation, with milder forms potentially identifiable later in pregnancy if fractures or deformities develop [35] (Table 1).

**Table 1.** Overview of COL1A1 gene and the pathologies involved.

| Study Focus | Key Findings | Citations |
| --- | --- | --- |
| Collagen in Disease Pathogenesis | Excessive collagen accumulation in the ECM is implicated in various diseases, including its role in cancer, by affecting cell migration and signaling pathways. | [20] |
| COL1A1 and Medical Conditions | The Sp1 binding site polymorphism in COL1A1 is linked to increased risks of ligament ruptures and other disorders like osteogenesis imperfecta and Ehlers–Danlos syndrome. The same polymorphism is associated with various multifactorial disorders. | [23] |
| Role of COL1A1 in ACL Ruptures | The COL1A1 gene's connection to ACL ruptures is emphasized, with a specific substitution in the Sp1 binding site potentially leading to increased COL1A1 expression and influencing ligament strength. | [33] |
| Genetic Inheritance and Diagnosis of COL1A1/2-OI | The inheritance pattern of COL1A1/2-OI is autosomal dominant, with variability in simplex cases based on severity. Prenatal genetic testing is possible for at-risk pregnancies, and prenatal ultrasound can diagnose severe forms of OI. | [35] |

## 3.2. COL5A1

The COL5A1 gene is responsible for encoding the α1 chain of type V collagen, a minor fibrillar collagen present in ligaments, tendons, and various other tissues [38]. Type V collagen, which constitutes around 10% of the collagen in ligaments, plays a role in the structure and regulation of type I collagen fibril diameters by integrating into the core of type I collagen fibrils [39]. A significant finding relates to the CC genotype of the BstUI restriction fragment length polymorphism (RFLP) within the 3′-untranslated region (UTR) of the COL5A1 gene. This genotype was notably less common in individuals with chronic Achilles tendinopathy [40]. Given the compositional and structural similarities between tendons and ligaments, the COL5A1 gene emerges as a potential genetic risk factor for ACL (anterior cruciate ligament) ruptures [41].

Posthumus et al. reported that the CC genotype of the BstUI RFLP within the COL5A1 gene's 3′-UTR was significantly underrepresented in female participants with ACL ruptures, but this was not observed in male participants. The study also found no association between the DpnII RFLP variant and ACL ruptures in either gender. Additionally, female participants with ACL ruptures reported a higher family history of ligament injuries, and this family history was also associated with the BstUI genotype in females. A novel discovery of this study was that female participants possessing the CC genotype of the COL5A1 BstUI RFLP appeared to have a reduced risk of ACL ruptures [42–44].

The Achilles tendon (AT) is recognized as one of the tendons most susceptible to rupture, accounting for approximately 20% of all large tendon injuries. This incidence is on the rise, as noted in studies [45,46]. The demographic most affected includes males, predominantly within the 40–50 age bracket. Such ruptures are significant, often marking the premature end of professional athletic careers. A notable characteristic of the AT is its midsection, which is the most common site for ruptures. This area, located 2 to 6 cm proximal to its insertion point, is particularly vulnerable due to its relatively poor vascularization [45]. Typically, the rupture is preceded by a degenerative process within the tendon. Eriksen et al. identified an increased presence of type III collagen at the rupture site, which they suggest may be a result of prior microtraumas and associated healing processes that reduce the tendon's tensile strength. These compromised tissue conditions could potentially hinder the healing process following a rupture [47].

The COL5A1 gene encodes the proα1(V) chain, essential in the assembly of type V collagen trimers; this collagen, being part of the fibrillar collagen subfamily, is predominantly found in the a1(V)2a2(V) isoform. It frequently co-assembles with type I collagen, forming heterotypic type I/V fibrils in tissues, including tendons, ligaments, bone, sclera, and cornea [48,49]. While the exact biological functions of type V collagen, especially in regulating type I collagen fibril diameter, are not fully understood, its significance in the development of connective tissues is well documented [50]. Notably, mutations in COL5A1 are linked to Ehlers–Danlos syndrome, characterized by the laxity and fragility of soft connective tissues [51]. Hence, variants in the COL5A1 gene, as identified in a South African study [52], could predispose individuals to increased risks of tendon or ligament injuries.

This association is further supported by Mokone et al. [52], who identified a significant correlation between the allelic variant marker rs12722 (BstUI RFLP) but not rs13946 (DpnII RFLP) in the 3′-UTR of the COL5A1 gene and symptomatic chronic Achilles tendinopathy in a South African cohort. Furthermore, September et al. confirmed the association of the BstUI RFLP (rs12722) with chronic Achilles tendinopathy in an Australian white population, pinpointing a critical region within the 3′-UTR of the COL5A1 gene that may increase susceptibility to this condition. These findings underscore the complex etiology of chronic Achilles tendinopathy, involving both genetic and non-genetic factors, and highlight a specific interval within the COL5A1 3′-UTR as a predisposing factor in both Australian and South African populations. Consequently, clinicians treating chronic Achilles tendinopathy should consider the potential role of genetic factors in assessing the risk of developing this injury [40,52] (Table 2).

### 3.3. COL11A1

Collagen type XI alpha 1 (COL11A1) is a component of type XI collagen, playing a vital role in bone development and collagen fiber assembly. Notably, COL11A1 expression is frequently upregulated in various cancers, with elevated levels correlating with adverse clinical outcomes, including poor survival, chemoresistance, and recurrence in numerous solid cancers. This upregulation suggests that COL11A1 contributes to tumor cell aggressiveness through multiple mechanisms. COL11A1, encoding one of the three alpha chains of type XI collagen, is primarily expressed in cartilage. In this tissue, it forms a heterotrimer with COL11A2 and COL2A1, essential for the assembly of type XI collagen [53,54]. Genetic mutations in the COL11A1 gene are linked to type II Stickler syndrome and Marshall syndrome. These autosomal dominant disorders are characterized by facial dysmorphism, myopia, and hearing loss [55,56]. Additionally, a specific single-nucleotide polymorphism in the COL11A1 gene has been associated with an increased risk of lumbar disc herniation [57]. In mice, a point mutation in the COL11A1 gene leads to the lethal chondrodysplasia phenotype, characterized by severe skeletal defects [58].

Recent research has elucidated that COL11A1, along with COL11A2 and the product of the COL2A1 gene, previously known as COL11A3, forms a heterotrimeric complex of collagen type XI [59,60]. This complex is crucial in regulating the fibrillogenesis of collagen types I and II. Besides its role in collagen formation, COL11A1 interacts with various other proteins, including ECM proteoglycans (e.g., biglycan, fibromodulin, and chondroadherin), ECM components (e.g., Thrombospondin-1, matrilin-1/3, chondrocalcin), and other collagen types (e.g., II, XI, XIV, XII, IX). Oncostatin M (OSM), an inflammatory cytokine, binds to collagen type XI in matrices derived from MDA-MB-231 breast cancer cells, further illustrating COL11A1's diverse interactions. Additionally, COL11A1 may form a unique heterotrimer with COL5A1 and COL5A2, although the precise stoichiometry of this association remains uncertain. Several studies have indicated that COL11A1, as well as COL11A2, can be associated with early-onset OA [61]. In a study conducted by Xu et al., they revealed that in homozygous cho/cho mice, the absence of the protein product of the Col11a1 gene is confirmed through Western blotting and immunohistochemical analysis. Given that type XI collagen is composed of three distinct polypeptide chains ($\alpha$1, $\alpha$2, and $\alpha$3), the formation of normal type XI collagen is likely hindered in the absence of the $\alpha$1 (XI) chain. This deficiency is posited to contribute to the cho phenotype, characterized by the lack of normal type XI collagen in cartilage. Furthermore, the cartilage in these mice is marked by unusually thick type II collagen fibrils. These findings indicate that a reduced level of type XI collagen in cartilage may be a primary factor in the onset of osteoarthritis (OA) [62]. This multifaceted role of COL11A1 in both normal physiological processes and pathologies like cancer and connective tissue disorders highlights its significance in cellular and molecular biology [63] (Table 2).

**Table 2.** Overview of COL5A1 and COL11A1 genes and the pathologies involved.

| Study Focus | Key Findings | Citations |
|---|---|---|
| COL5A1 Gene's Role in Tendons and Ligaments | The COL5A1 gene encodes the $\alpha$1 chain of type V collagen, crucial in ligament and tendon structure, and is involved in regulating type I collagen fibril diameters. | [38] |
| Gender-Specific Findings in ACL Ruptures | The CC genotype of the BstUI RFLP in the COL5A1 gene's 3'-UTR was underrepresented in females with ACL ruptures but not in males. No association was found between the DpnII RFLP variant and ACL ruptures in either gender. | [40,41] |
| Role of COL5A1 in Connective Tissue Disorders | Mutations in the COL5A1 gene are linked to Ehlers–Danlos syndrome. The gene is essential in forming collagen type V, which co-assembles with type I collagen in various tissues. | [52] |

**Table 2.** *Cont.*

| Study Focus | Key Findings | Citations |
|---|---|---|
| COL11A1 Gene in Bone Development and Cancer | The COL11A1 gene, a component of type XI collagen, is vital in bone development and is often upregulated in various cancers, correlating with adverse clinical outcomes. | [53,54] |
| Role of COL11A1 in Osteoarthritis Onset | In homozygous cho/cho mice, the lack of normal type XI collagen in cartilage due to a mutation in COL11A1 contributes to osteoarthritis, indicated by unusually thick type II collagen fibrils. | [62] |

## 4. The Meniscus: More Than Just Cartilage

The meniscus is integral to knee functionality, playing key roles in load transmission, shock absorption, and joint stability. However, the gene expression patterns in meniscal tears, especially concerning injury type and the influence of patient age and sex, remain largely unexplored. Meniscal fibrochondrocytes, a relatively sparse cell population, are responsible for synthesizing the meniscal matrix, which is primarily composed of water, collagen, and proteoglycans. Distinct from articular cartilage, which predominantly contains type II collagen, meniscal collagen is chiefly type I. The collagen fibers in the meniscus are mainly aligned circumferentially, supplemented by radial, perforating, and superficial fibers arranged more randomly. The proteoglycans within the meniscus, which are large hydrophilic molecules integrated into the collagen network, can absorb water up to 50 times their weight, although the meniscus has considerably less proteoglycan content compared to articular cartilage [5].

The cellular composition of the meniscus is more diverse than that of articular cartilage, which is primarily made up of a single cell type. The inner and middle regions of the meniscus primarily contain fibrochondrocytes—round or oval-shaped cells surrounded by a rich extracellular matrix. These cells predominantly produce type II collagen and are characterized by a lack of staining for cluster of differentiation (CD) 34, a marker typically associated with hematopoietic stem and progenitor cells. Additionally, the inner avascular region of the meniscus houses a unique progenitor cell population known for its multilineage differentiation potential and migratory activity [64,65]. In contrast, the outer region of the meniscus is mainly comprised of spindle-shaped fibroblast-like cells within a dense connective tissue matrix, primarily made up of type I collagen [66]. These cells exhibit positive staining for CD34 but do not stain for CD31, a marker for platelet endothelial cells. The presence of these fibroblast-like cells, along with vascularization, is thought to contribute to the healing capabilities of the outer meniscus. This complex cellular architecture underlines the meniscus's multifaceted role in knee joint function and its varied response to injury and repair processes [67].

Gene expression in meniscal tears demonstrates variability influenced by factors such as patient age, sex, and the nature of the injury. In younger patients, specifically those under forty, there is an increased expression of degradative enzymes and NFKB2, a transcription factor implicated in inflammatory and catabolic processes related to osteoarthritis. The presence of an accompanying ACL tear amplifies the expression of proinflammatory cytokines like IL-1ß and TNFa, chemokines such as CCL3 and CCL3L1, and the matrix-degrading enzyme MMP-13. Notably, in cases of concurrent meniscal and ACL injuries, there is a reduced expression of the matrix component aggrecan compared to isolated meniscal tears. These elevated levels of osteoarthritis-specific markers in younger patients with meniscal tears suggest a heightened catabolic (inflammatory) response, potentially indicating a predisposition to osteoarthritis development in these individuals, particularly following partial meniscectomy and ACL reconstruction [68,69].

Brophy et al.'s study corroborates these findings, revealing significantly higher expression levels of IL-1β, ADAMTS-5, MMP-1, MMP-9, MMP-13, and NFκB2 in patients under forty with meniscal tears compared to older individuals. In cases involving both meniscal and ACL tears in patients under forty, the expression of ADAMTS-4, ADAMTS-5, MMP-1,

and MMP-13 was also elevated. This group further exhibited significantly higher levels of IL-1β, TNFα, MMP-13, CCL3, and CCL3L1 and lower levels of aggrecan compared to those with only meniscal tears. The study also noted a sex-specific difference, with higher levels of CCL3L1 found in female patients of all ages with combined injuries. This intricate relationship between gene expression, age, injury pattern, and sex in meniscal tears underscores the complexity of the biological response to knee injuries and the potential implications for disease progression and treatment strategies [69] (Figure 1).

# Knee joint, arthritic

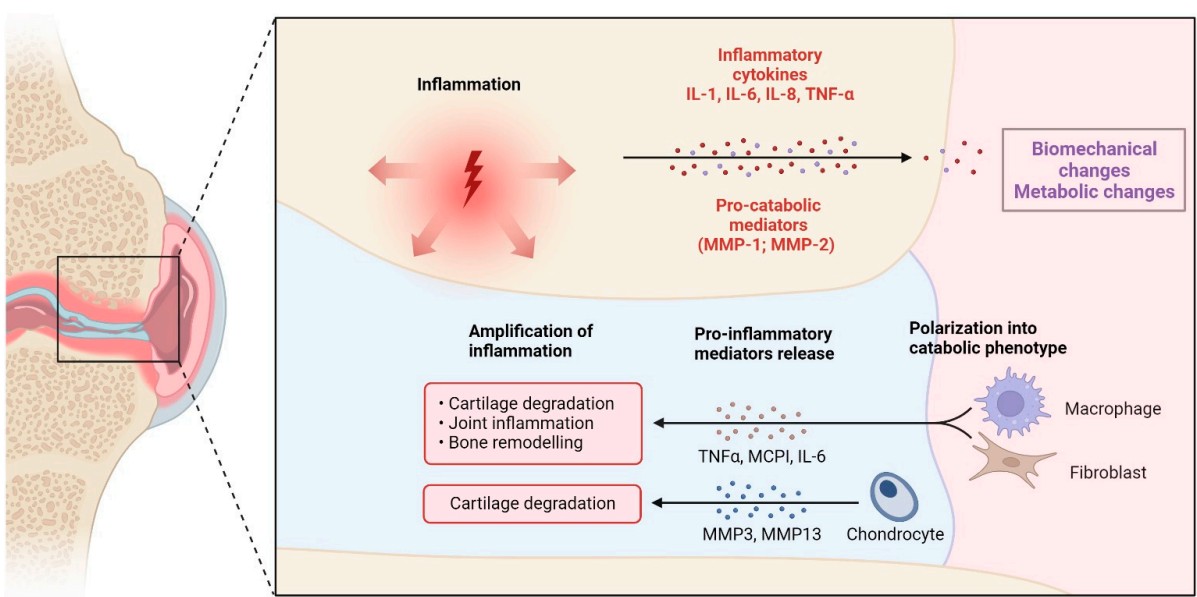

**Figure 1.** The pathophysiology of osteoarthritis (OA) at the knee joint encompasses a complex interplay of inflammatory processes and cartilage degradation. This condition is characterized by the progressive destruction of joint cartilage, a hallmark of OA. The underlying mechanism involves the release of proinflammatory cytokines, notably interleukin-1 (IL-1), IL-6, IL-8, and tumor necrosis factor-alpha (TNF-α). These cytokines play a pivotal role in mediating inflammation and joint damage. Additionally, OA pathology is exacerbated by the action of pro-catabolic mediators, including a family of enzymes known as matrix metalloproteinases (MMPs).

Knee ligament injuries represent a prevalent and severe category of sports-related traumas, substantially altering biomechanics. These injuries are classified into five types based on the affected ligament: anterior cruciate ligament (ACL), posterior cruciate ligament (PCL), transverse cruciate ligament (TCL), fibular collateral ligament (FCL), and patellar ligament (PL) injuries. It has been widely reported that delayed or inadequate treatment can lead to secondary complications, such as cartilage damage, meniscal injuries, and knee osteoarthritis (KOA). Ligament reconstruction is universally acknowledged as an effective remedial measure for restoring knee biomechanics, as evidenced in several studies [70–73]. Notably, ACL injuries constitute nearly half of these ligament injuries, often occurring in isolation [74].

In a systematic review conducted by Donelon et al., they studied the high-risk postures associated with ACL injuries as well as the training that practitioners should provide to the patient to reduce the risk of an ACL injury. The authors have deduced from their findings that the mitigation of KJLs (knee joint loads) can be achieved through a series of biomechanical adjustments: firstly, by decreasing the distances of lateral foot-plant, which in turn reduces the degree of hip abduction and aligns the foot closer to a neutral position, favoring a mid-foot or forefoot placement approach; secondly, by curtailing the angles and

motion of knee valgus and hip internal rotation at both initial contact (IC) and during the phase of weight acceptance (WA); thirdly, by eschewing lateral trunk flexion and striving to preserve a vertical trunk posture or inclining the trunk towards the intended movement direction; and lastly, by diminishing the magnitude of ground reaction forces (GRF) during WA, potentially through damping via enhanced knee flexion and by accentuating a more substantial portion of braking in the penultimate foot contact (PFC) [75].

According to Zhang et al., meniscal injuries, prevalent in both athletes and the general populace, frequently co-occur with traumatic ACL injuries, exacerbating stress and diminishing stability in the knee during extension and flexion. The onset of secondary conditions like cartilage wear and KOA is a significant risk if these injuries are not promptly addressed. Treatment strategies vary based on the severity of the injury and include conservative approaches, meniscus suturing, and meniscectomy [76,77]. Research by Magyar et al. [72] demonstrates that patients with meniscal injuries exhibit significantly reduced walking speed and knee range of motion (ROM) alongside increased cadence, step length, and duration of both support and double support phases. Conversely, Zhou [71] found no significant difference in maximum flexion and abduction–adduction angles between patients with meniscal injuries and healthy individuals, although the former exhibited larger minimum flexion angles and smaller maximum internal–external rotation angles during walking. Additionally, these patients experienced reduced stress areas and increased pressure in the knee while walking [76].

The gene expression profile in meniscus tears might provide crucial insights for orthopedic surgeons in deciding between resecting or repairing a tear. Traditional reluctance to repair degenerative tears stems from concerns about the structural integrity of the meniscus's longitudinal fibers and the limited blood supply to the inner meniscus. While tears in the vascularized periphery typically heal well, those in the less-vascularized inner meniscus often demonstrate poorer healing. Understanding the biology of these tears could be as crucial as other factors in predicting healing potential. Further research is needed to explore how differences in gene expression between degenerative and traumatic tear types may influence the healing of meniscus repairs [78,79].

Previous studies have shown limited evidence of a relationship between early degenerative changes in articular cartilage and gene expression in the injured meniscus. IL8, a cytokine induced in articular cartilage under mechanical, inflammatory, and metabolic stresses, is thought to play a significant role in osteoarthritis (OA). Osteoblasts from osteophytes increase IL8 production under nonphysiological loads, and elevated IL8 levels have been observed in the synovial fluid of total knee arthroplasty patients, correlating with the radiographic severity of OA. Meniscus cells from both healthy and OA knees increase IL8 production in response to proinflammatory stimulation, suggesting IL8's key role in the meniscus and other joint components [80–82].

Static compression of the meniscus has been associated with increased MMP1 expression, with MMP1 and MMP3 levels also being elevated in the synovial fluid of OA patients [83]. Animal studies indicate higher MMP1 and MMP3 expression in OA-afflicted knees, yet human studies show lower MMP3 levels in OA meniscus compared to controls, aligning with findings of higher MMP3 in traumatic versus degenerative meniscus tears. Additionally, MMP3 levels in synovial fluid correlate with preoperative pain scores [82,84] (Figure 2).

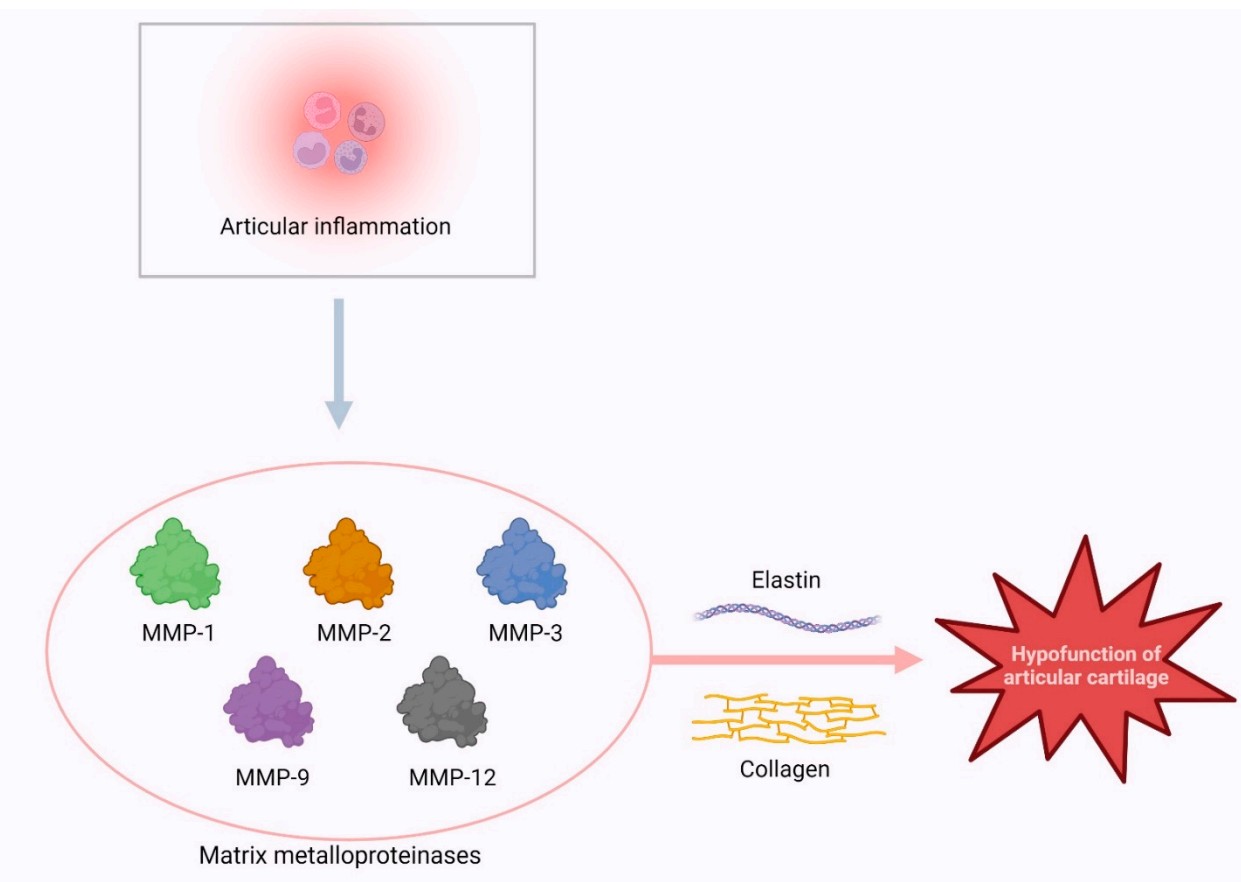

**Figure 2.** The matrix metalloproteinases effect on the hypofunction of the articular cartilage.

The systematic review and meta-analysis by Snoeker et al. [85] elucidated several prominent risk factors associated with meniscal tears, categorized into factors linked to degenerative and acute meniscal tears:

1. Age: Individuals aged over 60 years were identified as having a significantly increased risk for degenerative meniscal tears.
2. Gender: Male gender emerged as a notable risk factor for degenerative meniscal tears.
3. Occupational Activities: Work-related activities involving frequent kneeling and squatting were strongly associated with an elevated risk of degenerative meniscal tears.
4. Physical Activity: Climbing more than 30 flights of stairs was determined to be a significant risk factor for degenerative meniscal tears.
5. Sports Involvement: Active participation in high-impact sports such as soccer and rugby was identified as a key risk factor for acute meniscal tears.
6. Timing of ACL Surgery: A delay exceeding 12 months from the initial anterior cruciate ligament (ACL) injury to reconstructive surgery was found to significantly increase the risk of medial meniscal tears, but this correlation was not observed for lateral meniscal tears.

These findings highlight the varied etiological factors contributing to meniscal tears, underscoring the importance of considering age, gender, occupational and physical activities, sports participation, and the timing of surgical interventions in the assessment and management of meniscal injuries. The menisci play a crucial role in knee mechanics, but their contribution to joint health and the development of osteoarthritis (OA) through biological mechanisms beyond altered biomechanics remains less understood. According to Melrose et al. [86], there is a correlation between isolated meniscal tears or maceration and significant cartilage damage, suggesting a primary mechanical pathophysiological

role of the meniscus in OA. The scenario is more complex. Studies have shown that non-structural, intra-substance "degenerative" changes in the meniscus, as detected by MRI, do not necessarily correlate with cartilage pathology [87,88]. This finding indicates that while mechanical factors are critical, they may not fully explain the relationship between meniscal alterations and OA development. Additionally, it is important to note that meniscal tears and degeneration are frequently observed as "incidental" findings in knee MRIs. In these cases, these meniscal changes often do not correspond with OA pathology or symptoms [89] (Table 3).

**Table 3.** Overview of meniscus relevance and key findings regarding its pathologies.

| Study Focus | Key Findings | Citations |
|---|---|---|
| Meniscus Role in Knee Functionality | The meniscus is essential for load transmission, shock absorption, and joint stability. It is primarily made up of water, collagen (mostly type I), and proteoglycans. | [5] |
| Gene Expression in Meniscal Tears | Gene expression in meniscal tears varies by patient age, sex, and injury type. Younger patients show increased expression of degradative enzymes and proinflammatory cytokines, especially in cases of concurrent ACL injuries. | [68,69] |
| Knee Ligament Injuries and Meniscal Tears | Knee ligament injuries, particularly ACL injuries, can lead to secondary complications like meniscal tears and osteoarthritis. Proper treatment, like ligament reconstruction, is crucial. | [70–73] |
| Meniscus Tear Gene Expression and Repair Potential | The gene expression profile in meniscus tears may inform surgical decisions. Healing potential varies by tear location due to differences in blood supply and structural integrity. | [78,79] |
| Relationship Between Meniscus and OA | Meniscal tears and degeneration are linked to osteoarthritis development, but the relationship is complex and influenced by mechanical and biological factors. Studies highlight the need to consider both mechanical and non-structural changes in the meniscus when assessing OA risk. | [68,69] |

## 5. The Genetic Landscape of Knee Osteoarthritis

The knee joint is commonly affected by osteoarthritis (OA), a complex degenerative disease that often manifests clinically by middle age or even earlier. OA is characterized not only by the degeneration of cartilage, leading to pain, but also by alterations in the joint and surrounding tissues. There is a growing body of evidence suggesting that the progression of knee OA is frequently driven by biomechanical forces. The response of joint tissues to these forces can result in structural deterioration, symptomatic knee conditions, and diminished functional capacity. Prominent biomechanical risk factors contributing to the progression of OA include joint malalignment and meniscal tears. Despite the recognition of OA as a multifactorial condition, genetic factors have emerged as significant contributors to its development and progression. The genetic influence on OA is supported by various lines of evidence, including epidemiological studies highlighting family history and clustering, investigations involving twins, and the study of rare genetic disorders. This genetic predisposition indicates that OA's etiology extends beyond biomechanical and environmental factors, encompassing a complex interplay of genetic determinants. Understanding these genetic influences is crucial for developing more targeted and effective strategies for the prevention and management of OA [90].

Osteoarthritis (OA), a condition impacting synovial joints, exhibits a notable association between an increase in body mass index (BMI) and the risk of developing the disease, as highlighted by Geusens and van den Bergh [91]. This correlation underscores the need for a more comprehensive understanding of the genetic variances between weight-bearing joints (such as the knee, hip, and spine) and non-weight-bearing joints (including the hand, finger, and thumb).

Approximately 30–65% of the risk of developing osteoarthritis (OA) is attributed to genetic factors [92]. Warner et al.'s recent review [93] underscores significant findings from genetic association studies on OA, noting that genome-wide association scan (GWAS) studies have identified 21 independent susceptibility loci for OA. Since the publication of this review, further research has linked the single nucleotide polymorphism (SNP) rs4238326 in the ALDH1A2 gene with an increased risk of knee OA (KOA) in a Chinese cohort [94], expanding the gene's previously known association with hand OA in European populations [93].

Understanding OA also necessitates a focus on its epidemiology, which varies based on OA definitions, the specific joints examined, and the demographics of the population under study [92]. OA not only affects physical health but also has a substantial impact on mental health. Data from the Osteoarthritis Initiative (OAI) indicate that individuals with lower limb OA are more likely to develop depressive symptoms and exhibit greater odds of suicidal ideation compared to those without OA. Furthermore, OA is increasingly recognized as a risk factor for cardiovascular diseases, with a meta-analysis indicating a significantly higher risk of myocardial infarction among OA patients [95]. Other studies have found links between coronary heart disease and OA [96,97].

Age is a well-established risk factor for OA, with older individuals being more susceptible. Gender differences in OA prevalence are also notable, with women more likely to develop OA in the hand, foot, and knee but less likely to suffer from cervical spine OA. Racial disparities exist as well, with African-Americans being more prone to symptomatic knee and hip OA compared to other racial groups. Additionally, variations in radiographic OA features have been observed across different racial and ethnic groups [98].

Osteoarthritis (OA) is a multifactorial disease with a notable genetic component. Recent studies have emphasized the importance of genetics in OA by examining genetic variants and identifying various candidate genes associated with the disease. Among these, the GDF5 gene, which encodes for growth differentiation factor 5, stands out for its crucial role during the early stages of chondrogenesis. Abnormalities in the GDF5 gene manifest as a reduced size of the cartilage blastema, accompanied by alterations in the expression of cell surface molecules. This highlights the gene's pivotal role in the development and maintenance of joint health and its potential contribution to the pathogenesis of OA [99].

Growth differentiation factor 5 (GDF5), a bone morphogenetic protein, plays a significant role in joint formation and is expressed in various joint structures. Studies have demonstrated its efficacy in enhancing the healing of tendons, ligaments, and bones following trauma in mice [100,101].

A notable promoter polymorphism (rs143383) within the GDF5 gene has been identified as having a strong association with the development of both hip and knee osteoarthritis, particularly in Asian populations [101]. This polymorphism is one of the most consistently replicated genetic associations with knee osteoarthritis. A meta-analysis focusing on the rs143383 polymorphism in European cohorts has further confirmed the association of this functional variant with knee OA [102]. This variant was initially identified as having a significant correlation with knee and hip OA in Japanese and Han Chinese populations. The major (T) allele of this variant has been linked to a decreased expression level of GDF5, subsequently increasing the risk of OA [101].

Growth differentiation factor 5 (GDF5) plays a pivotal role in the development and maintenance of various bodily structures. In mice, Gdf5-null mutations result in a range of phenotypic abnormalities, including shorter feet and limbs, the absence of joints in digits, wrists, and ankles, altered tendon structures, missing knee ligaments, and a heightened risk of osteoarthritis [103–105]. In humans, mutations in the GDF5 gene are linked to a spectrum of clinical features such as short stature, shortened digits, joint dislocations or fusions, and hip- and knee-joint dysplasia, often accompanied by osteoarthritis [106,107].

Moreover, two common single nucleotide polymorphisms (SNPs) in the 5′ untranslated region (5′ UTR) of GDF5 (rs143383 and rs143384) have been identified as significant risk factors for osteoarthritis, particularly in Japanese and Chinese populations, increasing

the risk by approximately 1.3 to 1.8 times [101]. Functional studies revealed that the presence of T alleles at these SNPs leads to reduced expression of reporter genes in cultured articular cells and is associated with lower GDF5 transcript levels in the articular cartilage of individuals with osteoarthritis. Additionally, these 5′-UTR variants have been associated with variations in human height [108,109].

Wnt family member 1 (WNT1) and Wnt family member 10B (WNT10B) are integral components of the Wnt signaling pathway, which plays a critical role in the pathogenesis of osteoarthritis [110]. Notably, mutations in WNT10B have also been implicated in limb defects and dental anomalies [111]. Concurrently, WNT1 mutations have been associated with osteogenesis imperfecta.

The insulin-like growth factor 1 receptor (IGF1R) possesses tyrosine kinase activity and is pivotal in mediating the effects of insulin-like growth factor. It plays a significant role in regulating cartilage mineralization [112].

Nuclear receptor subfamily 3 group C member 1 (NR3C1) is responsible for encoding the glucocorticoid receptor (GR). The GR circulates in the cytoplasm and is actively involved in the inflammatory response [113]. In the context of osteoarthritis, the signaling of endogenous glucocorticoids within osteoblasts and chondrocytes has been shown to have deleterious effects [114].

The glucocorticoid receptor (GR) is encoded by the nuclear receptor subfamily 3 group C member 1 (NR3C1) gene, situated at the 5q31–32 locus. This gene undergoes alternative splicing, particularly of exon 9, resulting in two primary isoforms: GRα and GRβ. Additionally, alternative translation mechanisms lead to the generation of other GR isoforms. The presence of these various isoforms contributes significantly to the complexity of GR signaling. Furthermore, the regulation of GR is influenced by specific microRNAs (miRNAs), which can either inhibit the translation of GR transcripts or lead to their degradation. This additional layer of regulation by miRNAs adds further complexity to the post-transcriptional control of GR expression and function [115].

The single nucleotide polymorphism (SNP) rs1126464, a missense variant in the Dipeptidase 1 (DPEP1) gene, has been identified as a risk factor for osteoarthritis (OA). In a study utilizing data from the UK Biobank, this variant demonstrated a high posterior probability of causality (0.89) [116], and alterations in its expression have been observed in OA samples. The association of rs1126464 with OA susceptibility and clinical severity was further confirmed in the Chinese population through a combination of discovery and replication cohorts [117]. This variant is also considered a potential target for treatment and prognostic prediction in OA.

The majority of genetic variants associated with OA are located in non-coding regions of the genome, leading to the understanding that these variants predominantly affect gene regulation and protein expression rather than directly altering protein function. Epigenetic mechanisms, including DNA methylation, histone modifications, and non-coding regulatory RNAs, play a significant role in this process. These mechanisms can alter a cell's expression profile through chemical modifications, such as the methylation of cytosine–guanine dinucleotides (CpGs), or by influencing mRNA levels. Changes in DNA methylation patterns have been reported in human OA cartilage, and maintaining these methylation patterns is crucial for cartilage homeostasis [118–120]. Additionally, there is an established link between inflammation and the epigenetics of OA. Inflammatory mediators can impact the activity of enzymes responsible for DNA methylation and histone modifications, as well as affect the levels of non-coding RNAs [121].

There is an increasing consensus that hypertrophic differentiation of articular chondrocytes is a key process in the pathogenesis of osteoarthritis (OA), particularly in a subset of patients. This differentiation triggers a catabolic shift in the chondrocytes, irrespective of the initial cause. Factors such as IL-1β and biomechanical stimuli, like repetitive impulse loading, are known to induce this catabolic shift [122,123].

Leijten et al. have noted that the mRNA levels of GREM1, FRZB, and DKK1 are reduced in osteoarthritic cartilage [124]. These genes are more abundantly expressed in

articular cartilage compared to other types of hyaline cartilage and act as potent inhibitors of hypertrophic differentiation, establishing a strong correlation with OA. Their research also identifies a significant association between a genetic variant (SNP rs12593365) located in a genomic control region of GREM1 and radiographic osteoarthritis of the hip. This finding underscores the importance of understanding the interplay between transforming growth factor beta/BMP and WNT signaling pathways. These pathways are critically involved in the development of various tissues, including bone, cartilage, and intestinal epithelium [125].

Mutations in the Matrilin-3 gene (MATN3) have been identified as significant factors in the development of osteoarthritis, a progressive joint disease. Matrilin-3, a vital extracellular matrix protein, plays a crucial role in preserving the structural integrity and functional capacity of cartilage. Specific mutations within MATN3 have been found to alter the structure and function of Matrilin-3, leading to compromised assembly and organization of the cartilage extracellular matrix. Such disruptions diminish the cartilage's ability to endure mechanical stress and maintain optimal joint function [126].

The exact mechanisms by which Matrilin-3 mutations contribute to the pathogenesis of osteoarthritis are still under investigation. It is hypothesized that these mutations may interfere with the interactions between Matrilin-3 and other cartilage components, disrupt signaling pathways critical for cartilage homeostasis, and potentially exacerbate cartilage degradation and inflammatory responses within the joint. Matrilin-3 mutations have been particularly associated with early-onset osteoarthritis, typically affecting the hands, knees, and spine. Individuals with these mutations are likely to experience accelerated cartilage degradation, leading to an increased susceptibility to developing osteoarthritis at a younger age. Understanding the role of MATN3 mutations in osteoarthritis is crucial for developing targeted therapies and interventions to mitigate the progression of this debilitating condition [127].

To enhance the discovery of functionally relevant genes in osteoarthritis (OA), Butterfield et al. devised a comprehensive imaging method specifically designed to assess the mouse knee joint and its alterations related to the disease [128].

There are three distinct imaging modalities:

1. Iodine–contrast–enhancer (ICE) micro-computed tomography (µCT): This technique is employed to assess various aspects of cartilage and bone, including volume, thickness, and density.
2. Joint Surface Replication (JSR): JSR is utilized to quantify the extent of cartilage surface damage, providing a detailed evaluation of the joint's structural integrity.
3. Subchondral Bone X-ray Microradiography (scXRM): This method is used to determine the mineral density of the subchondral bone, which is crucial for understanding changes in bone quality associated with OA.

To establish a baseline for comparison, Butterfield et al. used 100 wild-type (WT) mice to generate a reference dataset representing a phenotypically normal joint. These baseline data were then validated and compared with joints that underwent osteoarthritis induction via destabilization of the medial meniscus (DMM) surgery. This approach allows for a detailed analysis of the structural changes in the knee joint associated with OA, providing valuable insights into the disease's progression and potential therapeutic targets [128].

The intricate biological complexity of osteoarthritis (OA) presents significant challenges, yet recent advances in molecular genetics offer promising avenues for the development of prognostic markers and customized therapeutics [129]. Notably, several disease-modifying osteoarthritis drugs (DMOADs) currently undergoing clinical trials are targeting proteins encoded by genes identified through genome-wide association studies (GWAS) [116,130]. These DMOADs include intra-articular therapies involving TGF-b and FGF18 growth factors, as well as Wnt inhibitors.

The intersection of OA genetics and epigenetics introduces the potential for epigenetic therapies. Many OA risk loci are associated with genes encoding histone-modifying proteins [131–133]. Histone deacetylase (HDAC) inhibitors have demonstrated efficacy in

reducing the expression of catabolic molecules like matrix metalloproteinases (MMPs) and IL-1 in OA chondrocytes and mouse models [134].

The use of CRISPR-Cas9 technology for epigenome editing is particularly promising. For instance, a dCas9-TET1 construct was employed to demethylate a hypermethylated mQTL within the RWDD2B promoter. This intervention increased RWDD2B expression, effectively counteracting the genetic risk associated with this OA locus. While this study was conducted in an immortalized cell line, it underscores the potential of epigenome editing in altering gene expression linked to disease risk. The development and implementation of DMOADs and epigenetic therapies in OA will require careful consideration of delivery methods into joint tissue, ensuring therapeutic permanence, and identifying suitable patient populations for specific treatments [135] (Table 4).

**Table 4.** Overview of OA biomechanical and genetic factors.

| Study Focus | Key Findings | Citations |
|---|---|---|
| OA's Biomechanical and Genetic Factors | OA is a complex degenerative disease influenced by biomechanical forces and genetic factors. Joint malalignment and meniscal tears are significant biomechanical contributors, while genetic predisposition is also a key factor. | [90] |
| Genetic Factors in OA | Genetic association studies, including GWAS, have identified various susceptibility loci for OA. The SNP rs4238326 in the ALDH1A2 gene is associated with an increased risk of knee OA. | [93,94] |
| GDF5 Gene's Role in OA | The GDF5 gene, crucial in chondrogenesis, is associated with OA. Abnormalities in GDF5 affect cartilage development and joint health. Promoter polymorphism rs143383 in GDF5 is strongly linked to hip and knee OA, particularly in Asian populations. | [101] |
| MATN3 Gene's Influence on OA | Mutations in the MATN3 gene, coding for Matrilin-3, affect cartilage integrity and contribute to early-onset OA. These mutations may disrupt cartilage homeostasis and exacerbate joint degradation. | [126] |
| Genetic and Epigenetic Research in OA | Advances in genetics and epigenetics offer insights for OA prognostic markers and therapeutics. DMOADs targeting proteins identified through GWAS are in clinical trials. Epigenetic therapies focus on histone modifications and non-coding RNAs. CRISPR-Cas9 technology for epigenome editing shows promise in altering gene expression related to OA risk. | [135] |

## 6. Conclusions

The study's exploration of genomic factors specific to knee joint functionality and pathology signifies a major leap in clinical orthopedics. It reveals how genetic predispositions affect susceptibility to knee injuries, joint degeneration, and recovery. Crucially, identifying key genes linked to collagen structure and joint stability enables the development of personalized treatment and prevention strategies for knee conditions. This research is particularly relevant in advancing the management of osteoarthritis, offering prospects for more precise, genetically tailored orthopedic treatments, thereby enhancing patient outcomes and quality of life.

The study's limitations include potential overgeneralization, complex genetic interactions, and uncertain clinical correlations. The findings may not apply universally due to varying genetic influences across populations. Conversely, its strengths lie in bridging genetics and biomechanics, offering insights into knee joint function and osteoarthritis. It identifies key genetic markers and suggests personalized rehabilitation strategies, enhancing the potential for tailored orthopedic interventions and advancing injury prevention and treatment optimization.

**Author Contributions:** Conceptualization, G.-L.I. and C.C.; methodology, L.I. and A.-D.C.; software, A.-D.C.; validation, R.-A.C.-B. and M.I.G.P.; formal analysis, A.-D.C.; investigation, G.-L.I. and A.-D.C.; resources, L.I.; data curation, M.I.G.P.; writing—original draft preparation, G.-L.I. and L.I.; writing—review and editing, A.-D.C.; visualization, R.-A.C.-B.; supervision, C.C.; project administration, C.C. and G.-L.I.; funding acquisition, C.C. All authors have read and agreed to the published version of the manuscript.

**Funding:** This research received no external funding.

**Institutional Review Board Statement:** Not applicable.

**Informed Consent Statement:** Not applicable.

**Data Availability Statement:** All data are available online in libraries such as PubMed.

**Conflicts of Interest:** The authors declare no conflicts of interest.

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
