# Peer review of "Genomic Determinants of Knee Joint Biomechanics: An Exploration into the Molecular Basis of Locomotor Function, a Narrative Review"

_cimb, doi:10.3390/cimb46020079_

Round 1

Reviewer 1 Report

Comments and Suggestions for Authors

This review seeks to provide a comprehensive exploration of the molecular basis underlying knee joint locomotor function

L2: add the review title on the title. Narrative review?

L86: add the purpose of this study

L88: is this study a Narrative review? Describe your review type and method of review. How did you search the articles?

From 2. Collagen and the Knee’s Structural Integrity To 4. The Genetic Landscape of Knee Osteoarthritis

Add the tables of literature review.

In particular, clearly describe what you have recently learned and the issues that need to be resolved at this time.

L628: conclusion  is too long. At conclusion section, you should write the summary of your results and clinical implication based on your results. I

Author Response

Dear Reviewer,

First of all, we would like to express our sincere gratitude for the insightful and constructive feedback you provided. We are pleased to inform you that we have carefully reviewed each of your suggestions and have successfully implemented all the recommended changes.

To summarize, we have:

  1. Added the narrative review in the title as you suggested
  2. We have added the purpose of the study as well as the study design, you will find it under 2.Materials and Methods
  3. We have added the tables as you suggested
  4. We have written another conclusion more spot on

We believe that the modifications made in response to your recommendations will not only strengthen our current project but also guide our future work. Your expertise and thorough review have been instrumental in this process.

We have also added the other reviewers suggestions in our manuscript and we believe that you will find it in a better form.

Kind regards,

The collective of authors

Reviewer 2 Report

Comments and Suggestions for Authors

The authors Georgian-Longin Iacobescu, Loredana Iacobescu, Mihnea Ioan Gabriel Popa, Razvan-Adrian Covache-Busuioc, Antonio-Daniel Corlatescu and Catalin Cirstoiu have submitted a manuscript (ID: cimb-2827176) entitled “Genomic Determinants of Knee Joint Biomechanics: An Exploration into the Molecular Basis of Locomotor Function” to Current Issues in Molecular Biology – Molecular Medicine.

The draft seems to be a reliable review with convincing data. I am pretty sure that this review will be of interest for the readership of the target journal.

Unfortunately, this information is missing:

Author Contributions:

Funding:

Institutional Review Board Statement:

Informed Consent Statement:

Data Availability Statement:

Conflicts of Interest:

Why is the text-part between line 567 and line 606 highlighted by a higher font size and an underline?

Furthermore, a discussion of some important publications in the field is missing:

Etoundi AC, Semasinghe CL, Agrawal S, Dobner A and Jafari A (2021) Bio-Inspired Knee Joint: Trends in the Hardware Systems Development. Front. Robot. AI 8:613574.

doi: 10.3389/frobt.2021.613574

Donelon TA, Dos'Santos T, Pitchers G, Brown M, Jones PA. Biomechanical Determinants of Knee Joint Loads Associated with Increased Anterior Cruciate Ligament Loading During Cutting: A Systematic Review and Technical Framework. Sports Med Open. 2020 Nov 2;6(1):53. doi: 10.1186/s40798-020-00276-5. PMID: 33136207;

Zhang L, Liu G, Han B, Wang Z, Yan Y, Ma J, Wei P. Knee Joint Biomechanics in Physiological Conditions and How Pathologies Can Affect It: A Systematic Review. Appl Bionics Biomech. 2020 Apr 3;2020:7451683. doi: 10.1155/2020/7451683. PMID: 32322301; PMCID: PMC7160724.

Author Response

Dear Reviewer,

Firstly, we wish to express our profound appreciation for the insightful and beneficial feedback you have provided. We are delighted to report that, following an in-depth examination of each of your proposals, we have successfully integrated all the changes you recommended.

To summarize:

  1. We have completed the information regarding the authors contributions, funding, conflicts of interests etc.
  2. We don't know why the text between line 567 and line 606 was highlighted by a higher font size, it had no purpose that way so we modified it
  3. Thank you for the articles that you recommended, we have included them in our bibliography.
  4. We have also addressed the other reviewers comments

We hold a strong belief that the alterations made in light of your suggestions will not only fortify our present project but also provide insightful direction for our upcoming initiatives. The depth of your expertise and the thoroughness of your review have been fundamental to our progress.

Thank you for all the kind suggestions and for your support!

Kind regards,

The collective of authors

Reviewer 3 Report

Comments and Suggestions for Authors

Dear Authors, 

I was pleased to review the paper entitled " Genomic Determinants of Knee Joint Biomechanics: An Exploration into the Molecular Basis of Locomotor Function" - 

- MDPI –

The present paper is very interesting, it focuses on a relevant clinical scenario, for orthopedics, potentially influencing the surgical and clinical practice for the management of knee joint. 

Thus, there are some minor remarks:

- Title: The title gives a fine idea of the topic to be covered

- Abstract: well written

- Introduction: I suggest concluding with the study aim and including some lines to the possible future role of your analysis.

- 2. Collagen and the Knee’s Structural Integrity: your work is very long, don't add details of oncological pathologies, lines 112-119.

            Adhere to editorial lines, lines 140-170.

            COL5A1: Reduce information regarding the Achilles tendon, not the subject of the study.

            COL11A1: what is its role in the knee joint?

- 3. The Meniscus: More Than Just Cartilage: why did you include “Snoeker et al. [75]”, is it linked to genomic determinants? If not, please remove.

- 4. The Genetic Landscape of Knee Osteoarthritis: Adhere to editorial lines, lines 567-606.

- The shoulder joint has been studied a lot from a Biomechanical point of view with sensors. This has allowed differentiation of diseases and thus treatments. Do you think there might be a role between measured biomechanics and genetics? (you could add doi: 10.5312/WJO.V12.I12.991).

- Include the limitations, strengths, and future implications of the work.

- The paper in general is well written, but some parts seem more like a book treatise with miscellaneous information than a paper with precise genetic topic. Please authors to streamline the manuscript, removing parts not strictly related to the subject.

The paper needs only minor changes.

Comments on the Quality of English Language

Minor editing of English language required

Author Response

Dear Reviewer,

We would like to start by extending our heartfelt thanks for the valuable and constructive feedback you have shared with us. It is with great pleasure that we inform you that after a thorough review of each suggestion you offered, we have effectively incorporated all the recommended changes.

To summarize, we have:

  1. After the introduction we have added a Materials and Methods section where you can find the study design as well as the objectives
  2. We have reduced as much as we could the information regarding the Achilles tendon, if you suggest removing more please let us know
  3. We have added some important information regarding the role of COL11A1 gene in the knee joint, to make it more clear
  4. Snoeker et al [75] is not linked to genomic determinants, however we believe that including that information is very relevant in the respective context.
  5. Thank you for the recommendation regarding the shoulder joint. This is a great idea for a further and shoulder concentrated study. We have read the suggested article and we have added to our bibliography
  6. We have modified the conclusion where we have added the limitations, strengths, and future implications of the work as you suggested
  7. We would like to mention that we have also addressed the other reviewers comments 

We are confident that the changes implemented based on your recommendations will enhance  the quality of our current project and also serve as valuable guidance for our future endeavors. Your expertise and detailed review have played a crucial role in this development.

Thank you for all your support!

Kind regards,

The collective of authors

Round 2

Reviewer 1 Report

Comments and Suggestions for Authors

Thank you for revision.

No comments.

Reviewer 2 Report

Comments and Suggestions for Authors

The authors of the manuscript entitled: “Genomic Determinants of Knee Joint Biomechanics: An Exploration into the Molecular Basis of Locomotor Function” have answered all questions of the reviewers accordingly. The manuscript is now ready for publication in Current Issues in Molecular Biology.